# Real-Time Monitoring of Metabolism during Exercise by Exhaled Breath

**DOI:** 10.3390/metabo11120856

**Published:** 2021-12-08

**Authors:** Martin Osswald, Dario Kohlbrenner, Nora Nowak, Jörg Spörri, Pablo Sinues, David Nieman, Noriane Andrina Sievi, Johannes Scherr, Malcolm Kohler

**Affiliations:** 1Faculty of Medicine, University of Zürich, 8008 Zürich, Switzerland; martin.osswald@usz.ch (M.O.); Dario.Kohlbrenner@usz.ch (D.K.); 2Department of Pulmonology, University Hospital Zürich, 8091 Zürich, Switzerland; Noriane.Sievi@usz.ch; 3ETH Zürich, Department of Chemistry and Applied Biosciences, 8049 Zürich, Switzerland; nora.nowak@org.chem.ethz.ch; 4Sports Medical Research Group, Department of Orthopaedics, Balgrist University Hospital, University of Zürich, 8008 Zürich, Switzerland; Joerg.Spoerri@balgrist.ch; 5University Centre for Prevention and Sports Medicine, Department of Orthopaedics, Balgrist University Hospital, University of Zürich, 8008 Zürich, Switzerland; 6University Children’s Hospital Basel, 4056 Basel, Switzerland; pablo.sinues@ukbb.ch; 7Department of Biomedical Engineering, University of Basel, 4123 Allschwil, Switzerland; 8Human Performance Laboratory, North Carolina Research Campus, Appalachian State University, Kannapolis, NC 28081, USA; niemandc@appstate.edu

**Keywords:** breath tests, metabolomics, exercise testing, physical activity

## Abstract

Continuous monitoring of metabolites in exhaled breath has recently been introduced as an advanced method to allow non-invasive real-time monitoring of metabolite shifts during rest and acute exercise bouts. The purpose of this study was to continuously measure metabolites in exhaled breath samples during a graded cycle ergometry cardiopulmonary exercise test (CPET), using secondary electrospray high resolution mass spectrometry (SESI-HRMS). We also sought to advance the research area of exercise metabolomics by comparing metabolite shifts in exhaled breath samples with recently published data on plasma metabolite shifts during CPET. We measured exhaled metabolites using SESI-HRMS during spiroergometry (ramp protocol) on a bicycle ergometer. Real-time monitoring through gas analysis enabled us to collect high-resolution data on metabolite shifts from rest to voluntary exhaustion. Thirteen subjects participated in this study (7 female). Median age was 30 years and median peak oxygen uptake (VO_2_max) was 50 mL·/min/kg. Significant changes in metabolites (*n* = 33) from several metabolic pathways occurred during the incremental exercise bout. Decreases in exhaled breath metabolites were measured in glyoxylate and dicarboxylate, tricarboxylic acid cycle (TCA), and tryptophan metabolic pathways during graded exercise. This exploratory study showed that selected metabolite shifts could be monitored continuously and non-invasively through exhaled breath, using SESI-HRMS. Future studies should focus on the best types of metabolites to monitor from exhaled breath during exercise and related sources and underlying mechanisms.

## 1. Introduction

The health benefits of regular exercise are well known and widely accepted. For example, regular exercise is effective in both preventing and treating cardiovascular, musculoskeletal, and metabolic diseases [1,2,3,4].

The physiological responses to acute exercise bouts have been studied extensively for a wide variety of exercise modes and workloads, but the underlying molecular mechanisms have not been fully described [5,6]. Muscles need to be supplied with energy from substrates via different pathways within a short time after initiating exercise. Carbohydrate, lipid, and protein substrate mobilization and utilization are precisely regulated to match the intensity and duration of exercise. For example, high energy demands during vigorous exercise bouts are met by increased utilization of intramuscular glycogen, and have typically been monitored with indirect calorimetry using ratios of oxygen uptake and carbon dioxide release, or in the blood by the increase in lactate concentration [7,8].

Advances in mass spectrometry technology now allow the simultaneous monitoring of hundreds of metabolites from blood, urine, and saliva samples periodically collected before, during, and after exercise. The rapidly expanding research area of exercise metabolomics has greatly improved our understanding of exercise-induced biological processes. During short-term graded exercise tests leading to peak oxygen uptake levels, for example, shifts in hundreds of plasma metabolites across multiple biochemical pathways have been reported, reflecting heightened anaerobic and aerobic metabolism including fatty acid oxidation, glycolysis products, release of Krebs cycle intermediates, inflammation, and oxidative stress [9,10].

Continuous monitoring of metabolites in exhaled breath has recently been introduced as an advanced method to allow non-invasive real-time monitoring of metabolite shifts during acute exercise bouts [11]. Little is known, however, about the types of metabolite that can be measured in expired breath samples during exercise [12,13], and how this profile differs when compared to metabolite shifts measured from blood samples [14,15,16,17]. Breath samples have been used previously for the monitoring of some volatile compounds using secondary electrospray high resolution mass spectrometry (SESI-HRMS) [18].

The purpose of this study was to measure metabolites continuously in exhaled breath samples with SESI-HRMS during a graded cycle ergometry cardiopulmonary exercise test (CPET). We also sought to advance the research area of exercise metabolomics by comparing metabolite shifts in exhaled breath samples with recently published data on plasma metabolite shifts during CPET [9,10].

## 2. Results

### 2.1. Participant Characteristics

Fourteen subjects (7 female and 7 male) were included in the study. Median (interquartile range (IQR)) age was 30 (27.0, 31.0) years, and median (IQR) maximum oxygen uptake (VO_2_max) was 50 (47.0, 53.0) mL/min/kg. One subject (male) had to stop the test due to psychological issues with the mask, and this subject was excluded from the analysis. The remaining thirteen subjects were healthy, non-smoking adults with regular physical activity/training. Ratings of perceived exertion (RPE) measured with the modified Borg scale indicated that all subjects reported voluntary exhaustion (a Borg value of 8–10) when the test was terminated. Furthermore, subjects had a significant increase in arterial blood lactate concentration during exercise testing, indicating maximal effort (see Appendix A). For further details regarding participant characteristics, see Table 1.

### 2.2. Metabolite Concentration of Several Pathways Decreased with Increasing Load

Significant changes were found in 32 out of 33 metabolites from several metabolic pathways, occurring between the start on the ramp and VO_2_max (Table 2, Figure 1A–D).

Decreases in exhaled breath metabolites were measured in glyoxylate and dicarboxylate (Figure 1A), tricarboxylic acid cycle (TCA) (Figure 1B), and tryptophan metabolic pathways (Figure 1C) during graded exercise (q = 0.001). The most marked decreases during the exercise bout occurred in five metabolites from the TCA, and these decreases began early in the CPET bout. The ketone body acetoacetate decreased during graded exercise, with no significant change in hydroxybutyrate (Figure 1D). Figure 2 compares changes in lipid- and carbohydrate-related metabolites using line graphs from one sample study participant. For individual protocol and metabolite intensities over time, see Appendix A.

### 2.3. Metabolites of Glycolysis and Oxo Fatty Acids Showed Inverse Behavior to Ketone Bodies

Ketone bodies (acetoacetate (AcAc) and hydroxybutyrate (OHB)) showed a short-term increase in the recovery phase after VO^2^max was achieved, with concomitant decreases in carbohydrate-related metabolites and oxo fatty acids. Subjects with high exercise capacity (i.e., VO^2^max) showed a second increase of ketone bodies shortly after the first during recovery. In contrast to the first increase, the second increase was accompanied with an increase in carbohydrate-related metabolites and a decrease of oxo fatty acids.

### 2.4. Ketone Bodies Correlated with Performance

There was a strong negative correlation (r = −0.87, *p* = 0.0002) between the oxygen uptake at VO_2_max and the short-term increase of ketone bodies in the recovery phase. Subjects with a higher VO_2_max showed the first short-term increase of ketone bodies earlier in the recovery phase. Hexose intensity increased simultaneously and strongly with the second ketone body increase in the recovery phase. Intensities of the mass spectrometric monitoring reflect the Fourier transformation of the image of the current measured at the detector of the orbitrap in arbitrary units—not to be confused with the exercise intensity that reflects the power relative to the athlete.

### 2.5. Metabolic Aerobic Threshold was Associated with Decrease of Carbohydrates and Fatty Acids

After an initial increase in intensity, carbohydrate metabolites, saturated fatty acids, and oxo fatty acids began to decrease in the first half of the exercise testing (Figure 2). This decrease started at the point of the ventilatory aerobic threshold.

## 3. Discussion

The purpose of this study was to measure metabolites continuously in exhaled breath samples with SESI-HRMS during a graded cycle ergometry CPET in 13 trained young adults. A total of 33 metabolites from glyoxylate and dicarboxylate, tricarboxylic acid cycle (TCA), and tryptophan metabolic pathways were identified, and most of these decreased during the incremental exercise bout.

To our knowledge, this is the first metabolomics-based study that continuously monitored metabolite changes during graded exercise to voluntary exhaustion (CPET) and the immediate recovery period. Other studies have monitored plasma metabolite shifts in response to CPET using samples collected before and immediately after exercise, and then during short-term recovery [5,6,9,10,23]. However, the methods used were heterogeneous and hamper comparisons between different studies. The most recent studies using sophisticated mass spectrometry platforms reported increases and decreases in hundreds of plasma metabolites in response to CPET [9,10]. CPET-induced shifts included increases in TCA intermediates, pyruvate and lactate, acylcarnitines, numerous lipid-related metabolites, and metabolites related to immunity, inflammation, and oxidative stress. At the same time, decreases occurred in specific types of amino acids, bile acids, and gut-derived products such as hippurate. In general, shifts in plasma metabolites with CPET may reflect heightened anaerobic and aerobic metabolism. 

Metabolite concentration of several pathways decreased with increasing load.

Comparing metabolic pathways between the start and end of the exercise testing revealed a significant enrichment in glyoxylate and dicarboxylate metabolism, TCA cycle, and tryptophan metabolism, respectively. With increasing load, energy fuels a shift from free fatty acids and glucose in blood, to substrates in the muscle itself, e.g., triglycerides and glucagon [8]. This may be the reason for the load-dependent decrease of these metabolites in our measurements. An accelerated uptake of TCA intermediates or a shift of substrate delivery from circulation to intramuscular metabolites may explain the decrease in TCA intermediates in exhaled breath. In accordance with prior research, intracellular TCA intermediates’ reservoir of molecules (pool size) increases with exercise intensity [24]. The measured decrease in exhaled TCA intermediates may represent a shift from extracellular to intracellular TCA intermediates. Another explanation, supported by prior plasma measurements [25], is that changes in breathing behavior may cause the decrease rather than a cellular concentration gradient. The cause for the measured changes remains subject to further investigations. 

Tryptophan pathway metabolites decreased during exercise. Metabolites of this pathway are associated with energy homeostasis, anti-inflammatory functions, and neuroprotection [26]. However, the function of tryptophan metabolism and glyoxylate and decarboxylate metabolism during exercise remain unclear. Interpreting our results, we hypothesize that substrate changes with increasing exercise intensity are due to limited carbohydrate delivery and may be a sign of increasing cellular stress with consecutive adrenergic activation. A similar interaction has been seen in the brain [27]. Fatty acid intensity remains close to constant after the initial increase. This supports the theory that transporter availability rather than adipose tissue catabolism is the limiting factor for fatty acid energy production [28]. For central carbon metabolites, several transporters support crossing membranes [29,30,31]. Whether the measured decrease in carbohydrate intensity is due to an increased uptake or a limited carbohydrate supply to the liver remains to be addressed in future research. However, since the glycogen storage capacity of the liver is on average 80 g (equivalent to an energy amount of 320 kcal) [32], the theory of limited carbohydrate availability seems to be more likely.

Metabolites of glycolysis and oxo fatty acids showed inverse behavior to ketone bodies.

Ketone bodies (i.e., acetoacetate (AcAc) and hydroxybutyrate (OHB)) showed a short-term increase in the recovery phase. The role of ketone bodies as muscular energy sources remains unclear [33]. A recognized function of ketone body metabolism is the spill-over pathway from beta-oxidation [34]. Since ketone bodies need to be transformed back to acetyl-CoA before they can be used as energy substrate, they are an unlikely alternative energy source in anaerobic metabolism. However, beta-hydroxybutyrate has several regulatory functions. It lowers metabolic rate and reduces sympathetic tone [35], and glycolysis alters glycolytic activity at the level of pyruvate dehydrogenase and phosphofructokinase [36]. Furthermore, beta-hydroxybutyrate activates, for example, c-Myc, MyoD, and tumor suppressor gene p53 by inhibition of histone deacylases, which in turn causes, inter alia, an increase in their expression [37]. The increase of ketone bodies shown in the recovery phase could, therefore, have a regulatory purpose. It may be a signal to down-regulate metabolism after a high-intensity exercise.

Simultaneously with the short-term ketone body increase, carbohydrate metabolites decreased. During exercise, catecholamines increase [38] and may therefore cause the changes shown in glucose levels [39]. 

Oxo fatty acids originate from the omega-oxidation pathway. This subsidiary pathway to beta-oxidation has a role in diabetes and starvation [40]. Furthermore, it has been proposed that dicarboxylic acids produced in the omega-oxidation pathway can be metabolized more efficiently in beta-oxidation [40]. The simultaneous increase of ketone bodies with the decrease in oxo fatty acids is plausible in view of the role of fatty acids as precursor of ketone bodies. The change may be due to the higher oxygen availability in th recovery phase and, therefore, increase of fatty acid oxidation. Whether these changes and the changes in carbohydrates are directly regulated by oxygen availability or connected to regulatory hormones, as well as a potential immediate regulatory effect of beta-hydroxybutyrate, remains to be unraveled. With regard to CPET-induced shifts in metabolites measured continuously from exhaled breath samples, much remains to be investigated. In this exploratory study, 33 metabolites were identified, and most decreased during the CPET bout. For example, strong decreases were measured in exhaled breath during CPET for five TCA-related metabolites in contrast to increases typically reported for these metabolites in plasma samples. Some of the metabolites identified in exhaled breath during exercise (e.g., the aromatic acid anthranilate, and the carboxylic acid 2-oxoadipate) have not been reported previously in plasma samples. Modest decreases in glycerol, two fatty acids (myristic and lauric acids), and acetoacetate from exhaled breath samples following CPET differed from increases measured for many lipid-related metabolites in plasma samples [9,10,28,41,42,43].

We hypothesised that measured intensities correlate linearly with exhaled metabolite concentration. A linear correlation between slightly volatile metabolite concentrations in blood and exhaled breath samples has been reported [19,44]. The contrasting pre-to-post-exercise patterns in metabolites measured in breath versus plasma samples cannot be easily explained, and will require additional research. One potential explanation is that during CPET, ventilation rates increase and thus dilute the concentrations of low-volatility metabolites that are being translocated from the circulation into exhaled breath samples [26]. Another potential explanation is the change in salivary composition with exercise intensity [12,45]. Two ketone bodies, acetoacetate and hydroxybutyrate, showed a short-term increase in the recovery phase when ventilation rates dropped quickly, perhaps reflecting that rates of production and transference to exhaled air outstripped disappearance rates.

Human breath contains many types of endogenous volatile organic compounds (VOCs), and the non-invasive detection of these trace gases (breathomics) has the potential for use in monitoring disease, medication responses, environmental exposure, and changes in metabolic and physiological function [13,46]. Breath isoprene, for example, may emerge as a clinically important breath metabolite, but its putative metabolic origin is still being debated, hindering its acceptance as a diagnostic marker. King et al. [13] showed that during 15 min of moderate cycling with breath-by-breath monitoring of VOCs, both isoprene and methyl acetate in exhaled breath increased sharply within the first minute of exercise. However, due to the negative ionization used in our study we are not able to measure acetone and isoprene and compare the results. These types of VOCs may be sensitive indicators of fluctuations in blood and respiratory flow rates, but the underlying physiological sources and mechanisms remain speculative.

## 4. Materials and Methods

### 4.1. Study Design

This was an exploratory study that included a CPET combined with mass spectrometric breath monitoring. Fourteen trained, normal weight and healthy male and female young adult subjects were recruited. In order to standardize blood glucose content, subjects were asked to drink a carbohydrate drink (long energy, Sponser, Wollerau Switzerland) two hours before testing (1.5 g/kg body weight). Furthermore, subjects were asked to refrain from sports activities one day before testing, and from eating, drinking (except water), chewing gum, lozenge, brushing their teeth, and using facial cosmetics at least one hour prior to the test. We conducted this study in accordance with the ethical principles of the Declaration of Helsinki and the Business Administration System for Ethics Committees (BASEC 2019-0030). All participants provided written informed consent.

### 4.2. CPET and Spirometry

CPET was performed on a bicycle ergometer (Excalibur Sport, LODE, Groningen, The Netherlands) connected to MetaLyzer software (MetaLyzer 3B-R3, Cortex, Leipzip, Germany). Before the CPET started, all subjects performed a spirometry test. Subjects followed a CPET protocol adapted from the recommendations of Radtke, T. et al. [47], with a resting phase of 2 min, a warm-up phase of 3 min at 10 Watts, a ramp with continuously increasing load (female, 25 Watts per minute; male, 30 Watts per minute; cadence 80–90 per minute) until voluntary exhaustion, and a recovery phase of 5 min at 10 Watts. At the beginning of the CPET and right before the end of the ramp, arterial blood was taken to measure lactate with ABL90 Flex plus, (Radiometer, Denmark). Subjective exertion and dyspnea was assessed using an adapted Borg subjective assessment scale (0 = no dyspnea and no exhaustion responses, 10 = maximal dyspnea and exhaustion responses) [48].

### 4.3. SESI-HRMS

Subjects wore an adapted CPET facemask connected to both the turbine of the CPET and the SESI-HRMS. Air was sucked through an extended sampling line coated with SilcoNert 2000 (SilcoTek GmbH, Bad Homburg, Germany) to the secondary electrospray ionization source (SuperSESI, FIT, Madrid, Spain) and into the orbitrap mass spectrometer (QExative Plus, ThermoFisher, Waltham, MA, USA). The sampling line was heated to 130 °C to prevent adsorption or condensation. The spray solution consisted of 0.1% formic acid in water (LC-MS grade, Merck, Darmstadt, Germany). We used 20 µm diameter capillaries with a length of 50 cm (New Objectives, Littleton, CO, USA), and a starting pressure of 1.3 bar. The ion core was heated to 90 °C, exhaust flow rate was limited to 450 mL/min, and a current of 3.5 kV and no aux gas or sweep gas were used.

On the QExactive the following settings were used: resolution of 70,000 at 200 *m*/*z*, 1 micro scan, a range of 50–750 *m*/*z*, automatic gain control target 1 × 10^6^, maximum injection time 200 ms; and the ion transfer capillary was heated to 275 °C.

The monitoring frequency was three scans per second. The sample flow was determined by the vacuum pump and was approximately 1 L per minute. Acquisition was started and stopped simultaneously to CPET. 

### 4.4. Data Analysis

The raw mass spectrometric data were converted to the open format *.mzXML by MSConvert (ProteoWizard) [49]. For the processing of the data, the functions “mzxmlread”, “mzxml2peaks” and “mspeaks” in Matlab R2020a (MathWorks) were used. The summed intensities of each peak within full width half maximum were used for further analysis. Exhalations were extracted using a homemade algorithm based on the feature *m*/*z* 60.9931, which showed reproducible behavior of the exhalations.

CPET data was exported to *.xlsx files. The continuous data was interpolated with “pchip” to steps of 0.3 s and matched to the closest time point of the mass spectrometric data. The key values (ventilatory threshold 1, ventilatory threshold 2, etc.) were set by blinded experts, imported, and used for further calculations. For time series cluster analysis, features with less than 50% zero values (present in at least 80% of the subjects) were included. The normalized moving mean of the time series of features were clustered using “clusterdata” algorithm. Functional analysis was done using mummichog algorithm of Metaboanalyst 5.0 [50]. For enrichment analysis, q-values were calculated from p-values with “fdr_bh” based on 659 features detected in negative ionization mass spectrometry [51]. *p*-values were calculated with Wilcoxon signed rank test by pairwise comparisons of the area under the curve within 50 scans at the start on the ramp and at VO^2^max. To determine significance in boxplots, q-values were calculated as mentioned above. Q-values of ≤0.001 were considered significant. In this study, we focused on negatively charged ions. 

Metabolites were identified by exact mass, enrichment analysis, and previously known exhaled metabolites.

## 5. Conclusions

This exploratory study showed that continuous monitoring of metabolites in exhaled air at high time resolution (three spectra per second) with SESI-HRMS during graded exercise was feasible. We were able to observe significant changes in glyoxylate and dicarboxylate, tricarboxylic acid cycle (TCA), and tryptophan metabolic pathways. Our findings suggest that continuous monitoring of metabolite shifts from exhaled air during exercise should focus on monitoring platforms dedicated to volatile and slightly volatile metabolites.

## Figures and Tables

**Figure 1 metabolites-11-00856-f001:**
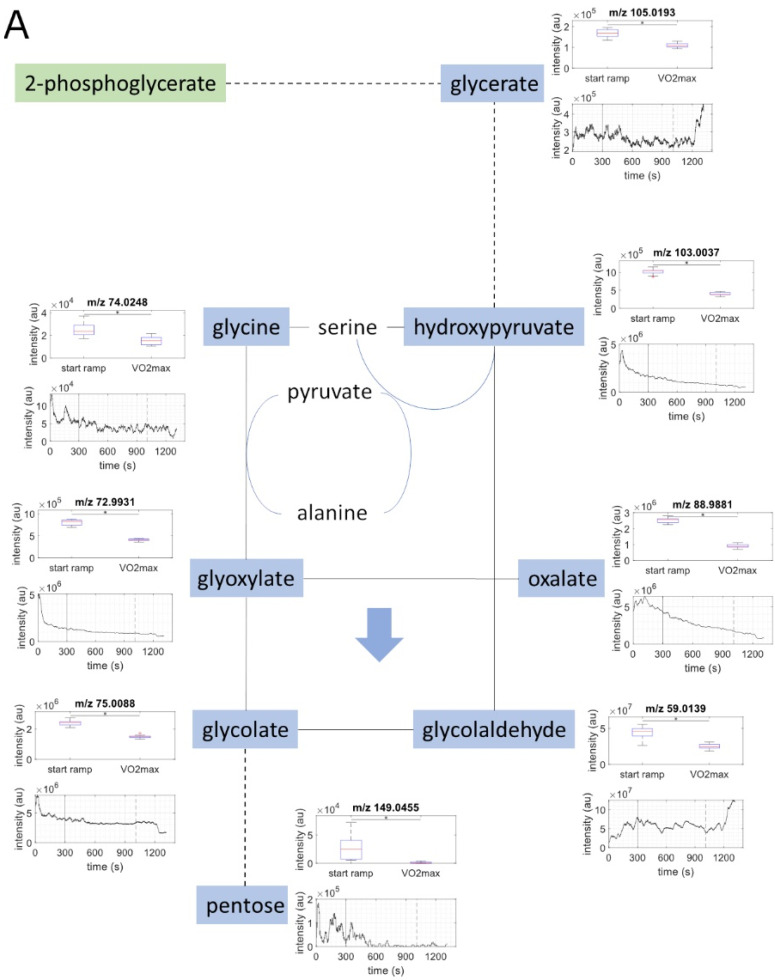
Metabolic pathways map. Metabolites that were identified in exhaled breath are marked with dark colour, while lighter colors are used for unidentified metabolites that demarcate important points in the pathways. Pathways including metabolites that decreased significantly between the start of the ramp and VO^2^max are: (**A**): blue for glyoxylate metabolism; (**B**): yellow for TCA cycle; and (**C**): grey-blue for tryptophan metabolism. Trends for changes in metabolite intensities are marked with arrows. Furthermore, (**D**): glycolysis is marked in green, lipid metabolism in brown, and ketone pathway in red. Summary data for each identified metabolite are represented with box plots (with median, quartiles, maxima and outliers) using intensities for all subjects from the start on the ramp to VO^2^max. Continuous intensity data over time are represented from one sample subject (subject 6).

**Figure 2 metabolites-11-00856-f002:**
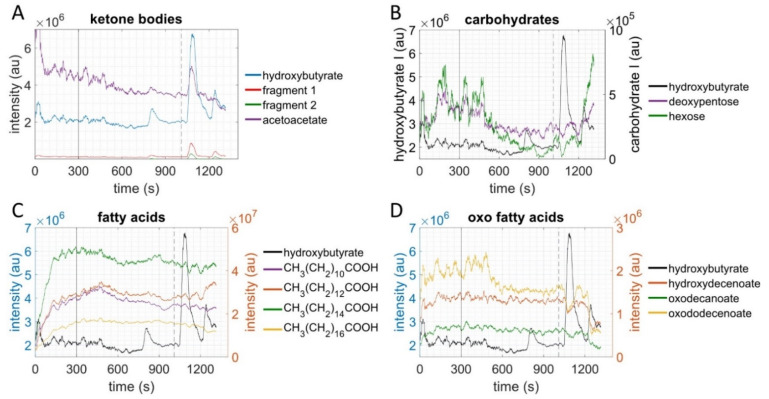
Intensity data over time from one sample subject (subject 6). Solid vertical lines represent the start on the ramp, dashed vertical lines represent VO^2^max. The left y-axis represents the ketone bodies in (**A**) and hydroxybutyrate in (**B**,**C**), respectively. The right y-axis represents the other metabolites. (**A**): Hydroxybutyrate (OHB), acetoacetate, and fragments; (**B**): OHB, hexose and deoxypentose; (**C**): OHB, stearate, palmitate, myristate and laurate; (**D**): OHB, hydroxydecenoate, oxodecanoate and oxodecenoate.

**Table 1 metabolites-11-00856-t001:** Participant characteristics.

Characteristic	Subjects (N = 13)
Sex = male (%)	6 (46.2)
Age (median [IQR]), years	30.0 [27.0, 31.0]
BMI (median [IQR]), kg/m^2^	22.8 [22.3, 24.0]
Weight (median [IQR]), kg	70.0 [62.5, 77.7]
Smoking = no (%)	13 (100.0)
Spirometry FVC ^1^ (median [IQR]), L	5.3 [4.3, 6.4]
FVC ^1^ % predicted (median [IQR])	124.0 [110.0, 128.0]
FEV1 ^2^ (median [IQR])	4.3 [3.7, 5.0]
FEV1 ^2^ % predicted (median [IQR])	110.0 [106.0, 120.0]
FEV1/FVC ^3^ (median [IQR])	81.0 [76.0, 84.0]
Exercise per Week Activity number (median [IQR])	4.5 [2.0, 6.0]
Activity duration (median [IQR]), min	150.0 [120.0, 210.0]
Activity intensity ^4^ = moderate (%)	13 (100.0)
Adapted Borg scale (0–10) Dyspnea start (median [IQR])	0.0 [0.0, 0.5]
Legs start (median [IQR])	1.0 [0.5, 1.0]
Dyspnea end (median [IQR])	9.5 [9.0, 10.0]
Legs end (median [IQR])	9.0 [7.5, 9.8]
Cardiopulmonary exercise test HR ^5^ at VT1 ^6^ (median [IQR]), beats per min	141.0 [123.0, 144.0]
HR ^5^ at VT2 ^7^ (median [IQR]), beats per min	168.5 [163.8, 179.0]
Maximal HR ^5^ (median [IQR]), beats per min	185.0 [183.0, 190.0]
Power at VT1 ^6^ (median [IQR]), W	138.0 [88.0, 190.0]
Power at VT2 ^7^ (median [IQR]), W	234.5 [213.5, 296.2]
Maximal Power (median [IQR]), W	290.0 [261.0, 366.0]
VO_2_max ^8^ (median [IQR]), mL/min/kg	50.0 [47.0, 53.0]

^1^ FVC = forced vital capacity, ^2^ FEV1 = forced expiratory volume in 1 s, ^3^ FEV1/FVC = ratio of forced vital capacity to forced expiratory volume in 1 s, ^4^ Activity intensity = percent with a median subjective moderate activity intensity, ^5^ HR = heart rate, ^6^ VT1 = ventilatory threshold 1, ^7^ VT2 = ventilatory threshold 2, ^8^ VO_2_max = maximum oxygen consumption.

**Table 2 metabolites-11-00856-t002:** Identification state of exhaled metabolites.

Sum Formula	Compound Name	Measured Mass (M-H) ^1^	Exact Mass (M-H) ^1^	Mass Error (ppm) ^2^	Identified Based on	Course of Metabolites during Exercise ^3^
C6H12O6	hexose	179.0562	179.0556	3.56	exact mass & pathway mapping, and Nowak, N. et al. [19]	↓
C5H10O4	deoxypentose	133.0507	133.0501	4.63	exact mass & pathway mapping	↓
C5H10O5	pentose	149.0455	149.0450	3.37	exact mass & pathway mapping, and Nowak, N. et al. [19]	↓
C3H4O3	pyruvate	87.0088	87.0082	6.68	exact mass & pathway mapping	↓
C2H4O2	glycolaldehyde	59.0139	59.0133	10.09	exact mass & pathway mapping	↓
C3H6O4	glycerate	105.0193	105.0188	4.92	exact mass & pathway mapping	↓
C3H4O4	hydroxypyruvate	103.0037	103.0031	5.5	exact mass & pathway mapping	↓
C2H2O3	glyoxylate	72.9931	72.9926	7.28	exact mass & pathway mapping	↓
C2H2O4	oxalate	88.9881	88.9875	6.93	exact mass & pathway mapping	↓
C2H3O3	glycolate	75.0088	75.0082	7.75	exact mass & pathway mapping	↓
C2H4NO2	glycine	74.0248	74.0242	8.06	exact mass & pathway mapping	↓
C4H6O3	acetoacetate /oxobutyric acid	101.0244	101.0239	5.26	exact mass & pathway mapping	↓
C4H8O3	hydroxybutyrate	103.0401	103.0395	5.64	exact mass & pathway mapping	Constant
C4H4O5	oxaloacetate	130.9986	130.9980	4.21	exact mass & pathway mapping, and Rioseras, A. et al. [20]	↓
C4H6O5	malate	133.0143	133.0137	4.52	exact mass & pathway mapping, and Rioseras, A. et al. [20]	↓
C4H4O4	fumarate	115.0037	115.0031	4.93	exact mass & pathway mapping, and Rioseras, A. et al. [20]	↓
C4H6O4	succinate	117.0194	117.0188	5.27	exact mass & pathway mapping, and Rioseras, A. et al. [20]	↓
C5H6O5	oxoglutarate	145.0142	145.0137	3.46	exact mass & pathway mapping, and Rioseras, A. et al. [20]	↓
C6H7NO4	2-aminomuconate	156.0302	156.0297	3.32	exact mass & pathway mapping	↓
C6H8O5	oxoadipate	159.0299	159.0293	3.47	exact mass & pathway mapping	↓
C8H7NO3	formylanthranilate	164.0353	164.0348	3.24	exact mass & pathway mapping	↓
C7H7NO2	anthranilate	136.0404	136.0399	4.02	exact mass & pathway mapping	↓
C6H7NO3	2-aminomuconate semialdehyde	140.0353	140.0348	3.8	exact mass & pathway mapping	↓
C7H7NO3	3-hydroxyanthranilate	152.0353	152.0348	3.5	exact mass & pathway mapping	↓
C6H5NO2	picolinate	122.0248	122.0242	4.89	exact mass & pathway mapping	↓
C3H8O3	glycerole	91.0401	91.03952	6.38	exact mass & pathway mapping	↓
C12H24O2	dodecanoic acid /lauric acid	199.1705	199.1698	3.49	exact mass & pathway mapping	↓
C14H28O2	tetradecanoic acid /myristic acid	227.2018	227.2011	3.06	exact mass & pathway mapping	↓
C18H36O2	octadecanoic acid	283.2643	283.2637	2.1	exact mass, and Anokhina, T.N. et al. [21]	↓
C16H32O2	hexadecanoic acid	255.2329	255.2324	1.94	exact mass, and Anokhina, T.N. et al. [21]	↓
C10H20O3	hyroxydecanoic acid	187.1341	187.1334	3.64	exact mass and Gaugg, M. et al. [22]	↓
C10H16O3	oxodecenoic acid	183.1028	183.1021	3.72	exact mass	↓
C12H20O3	oxododecenoic acid	211.1340	211.1334	2.75	exact mass	↓

^1^ (M-H)—deprotonated molecule; ^2^ ppm—parts per million; ^3^ courses between start on ramp and maximal ventilatory oxygen uptake.

## Data Availability

Data supporting reported results can be found on the openly available repository Zora.

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
