# Peer review of "Real-Time Monitoring of Metabolism during Exercise by Exhaled Breath"

_metabolites, 2021, doi:10.3390/metabo11120856_

Round 1

Reviewer 1 Report

The manuscript under-revision presents the results of a study analysing, during acute exercise, the time profiles of SESI-HRMS ion peaks corresponding to breath metabolites. The authors’ group has been engaged in this field for many years, as confirmed by the list of references, and has significant experience in the field of breath research. The text is clear and easy to understand. The data behind the presented results are obviously a result of huge amount of work. However, in my opinion, this article lacks some details and wider discussion of the meaning of the results. If and when this is added even the most discerning readers will enjoy reading it and the value of the manuscript would be increased.

The results of the study are in the form of comprehensive data including a large map showing the metabolic pathways alongside the associated time profiles of all targeted metabolites (Figure 1 and Supplementary materials). The data obtained in the present study are a valuable contribution in the field of breath metabolite monitoring during and after acute exercise bouts. Also, the study demonstrates the great potential of SESI-HRMS for the real-time monitoring of VOCs in gaseous samples including breath.

In my opinion, the manuscript can be accepted for publication in Metabolites journal after the authors have had a chance to consider the following points:

  • Figure 1 is very comprehensive although far too detailed. This clever figure contains the metabolic pathways including the m/z of product ions of selected compounds, their time profiles as well as their associated box-and-whiskers plots. The data as laid out in this figure may be hard to see in the printed journal and also, they are not easy to interpret without considerable additional explanation. A huge amount of information shown in Figure 1 makes the figure difficult to read, even in the electronic form (in the quality of the current PDF file). The idea of Figure 1 is really good, but it needs to be available as an A1 sheet (perhaps as a supplementary pdf). On the other hand, Table S1 is really useful and should be in the main text of the paper (perhaps with only the essential columns). The authors should consider to add symbols into this table indicating "increased", “decreased” and "constant" for each phase of the exercise or even a mean value of the targeted VOC relative ion signal compared to the initial signal at time 0.
  • The Figures and Table certainly require further discussion within the text of the manuscript. The discussion in the current form does not give sufficient credit to the data.
  • The use of the term "measurements" is not appropriate for this study. This word implies quantification of metabolites which was not done. For example, "Continuous measurement of metabolites in exhaled breath..." on line 61and elsewhere. More appropriate terms in place of “measurements” would include words such as “monitoring”, “analyzing”, “detecting” as appropriate at different sections. A clear and brief discussion of how the signal intensities presented in arbitrary units for the various m/z ions relate to the concentrations of the underlying compounds in the breath samples.
  • There are several studies available in the literature which have focused on the measurement of metabolites in exhaled breath during exercise. The work of King et al (ref. 23) is an important study which must be mentioned in the introduction and should be discussed in more detail later on in the manuscript within the discussion (e.g. similarities and differences comparing the protocol, methods, and results between the submitted manuscript and that of King et al). The study by Smith et al: "SIFT-MS Analysis of Nose-Exhaled Breath; Mouth Contamination and the Influence of Exercise. Current Analytical Chemistry, 2013, 9, 565-575" by which several metabolites were measured in exhaled breath during exercise on a stationary bicycle has been missed by the authors.
  • The discussion of the results lacks a comparison between the present data and the results of previously published studies. In addition, there was no mention of any observation of acetone (the major ketone body) or isoprene during the present study According to the mass range given in the manuscript, these compounds could be monitored during exercise, and if so they must be discussed. If not, this should be clearly mentioned and the corresponding reasons for this discussed.
  • Section “4.3. SESI-HRMS” should be extended by incorporating further details such as the breath sample flow; whether the breath sample was diluted by a certain flow of either nitrogen or dry air, other SuperSESI parameters; temperature of ion transfer capillary, the type of mass spectrometer (orbital trap), etc.

Author Response

  1. Figure 1 is very comprehensive although far too detailed. This clever figure contains the metabolic pathways including the m/z of product ions of selected compounds, their time profiles as well as their associated box-and-whiskers plots. The data as laid out in this figure may be hard to see in the printed journal and also, they are not easy to interpret without considerable additional explanation. A huge amount of information shown in Figure 1 makes the figure difficult to read, even in the electronic form (in the quality of the current PDF file). The idea of Figure 1 is really good, but it needs to be available as an A1 sheet (perhaps as a supplementary pdf). On the other hand, Table S1 is really useful and should be in the main text of the paper (perhaps with only the essential columns). The authors should consider to add symbols into this table indicating "increased", “decreased” and "constant" for each phase of the exercise or even a mean value of the targeted VOC relative ion signal compared to the initial signal at time 0.

Response to comment 1 of reviewer 1:

We are happy to take the table S1 into the main text if the editor wants us to do so. We have put this table into the supplement because we think that only few readers will be interested in the exact state of identification of the single metabolites.

We added the indication of “increased”, “continuous”, and “decreased” to the table S1

We agree that Figure 1 is too small to read in A4 format. Is it possible in this online journal to zoom the figures by e.g. clicking on the small version? Otherwise, we would need to cut the figure in several separate figures, containing only parts of the network, but we think this would reduce the clarity of the Figure and make it hard to read.

  1. The Figures and Table certainly require further discussion within the text of the manuscript. The discussion in the current form does not give sufficient credit to the data.

 Response to comment 2 of reviewer 1:

Thank you for the comment. We added an additional subchapter to the Discussion section discussing the dedicated data in more detail.

  1. The use of the term "measurements" is not appropriate for this study. This word implies quantification of metabolites which was not done. For example, "Continuous measurement of metabolites in exhaled breath..." on line 61and elsewhere. More appropriate terms in place of “measurements” would include words such as “monitoring”, “analyzing”, “detecting” as appropriate at different sections. A clear and brief discussion of how the signal intensities presented in arbitrary units for the various m/z ions relate to the concentrations of the underlying compounds in the breath samples.

Response to comment 3 of reviewer 1:

We thank the reviewer to point this out to us, we changed the wording accordingly in the revised version, furthermore we now discuss the relation between the mz-ions and the concentrations in the discussion.

  1. There are several studies available in the literature which have focused on the measurement of metabolites in exhaled breath during exercise. The work of King et al (ref. 23) is an important study which must be mentioned in the introduction and should be discussed in more detail later on in the manuscript within the discussion (e.g. similarities and differences comparing the protocol, methods, and results between the submitted manuscript and that of King et al). The study by Smith et al: "SIFT-MS Analysis of Nose-Exhaled Breath; Mouth Contamination and the Influence of Exercise. Current Analytical Chemistry, 2013, 9, 565-575" by which several metabolites were measured in exhaled breath during exercise on a stationary bicycle has been missed by the authors.

Response to Comment 4 of reviewer 1:

Thank you for making us aware of the interesting work by Smith et al., we included this paper in our Discussion section. Additionally, we  reference King et al. in the Introduction section as well.

  1. The discussion of the results lacks a comparison between the present data and the results of previously published studies. In addition, there was no mention of any observation of acetone (the major ketone body) or isoprene during the present study According to the mass range given in the manuscript, these compounds could be monitored during exercise, and if so they must be discussed. If not, this should be clearly mentioned and the corresponding reasons for this discussed.

Response to Comment 5 of reviewer 1:

We thank the reviewer for this comment. Our analysis was only done in negative ionization mode. This is because we wanted to focus on the spectrum of metabolites well known from plasma studies . We clarified our Methods section to emphasize the method used. To obtain valid results of acetone, a positive mode analysis would be needed. Accordingly, we may not draw conclusions on acetone behavior in this work. However, changes in exhaled isoprene are mainly discussed with the onset of movement (e.g. King et al) or changes in positioning.

  1. Section “4.3. SESI-HRMS” should be extended by incorporating further details such as the breath sample flow; whether the breath sample was diluted by a certain flow of either nitrogen or dry air, other SuperSESI parameters; temperature of ion transfer capillary, the type of mass spectrometer (orbital trap), etc.

Response to Comment 6 of reviewer 1:

We thank the reviewer to point this out, We added the details in the dedicated paragraph.

Reviewer 2 Report

Dear Authors, the manuscript is well written and organised. I have a general question to authors: did you evaluate the impact of contamination of breath samples with saliva during analysis? How did u excluded it? It is well reported that salivary flow rate decrease over exercise and pH as well. It is possible to think a potential relationship between the concentration of metabolites and salivary pH.

Author Response

  1. Dear Authors, the manuscript is well written and organised. I have a general question to authors: did you evaluate the impact of contamination of breath samples with saliva during analysis? How did u excluded it? It is well reported that salivary flow rate decrease over exercise and pH as well. It is possible to think a potential relationship between the concentration of metabolites and salivary pH.

Response to comment 1 of reviewer 2

We thank the reviewer for this important comment. We  agree that contamination of breath samples could be a relevant source of exhaled breath metabolites and that results could be influenced by that. To emphasize this,  we added a statement on the possible influence of salivary composition to the limitations paragraph of the revised manuscript alongside a dedicated reference.

At the current state we do not exactly know to which parts exhaled metabolites originate from alveoli, lower and upper respiratory tract glands, stomach, etc. that all is part of exhaled breath.

Reviewer 3 Report

Metabolites-1441882

Osswald et al.

Summary

The investigators have applied the emerging approach of real-time “breathomics” analysis to a graded exercise model in humans, which is a potentially powerful means to identify quick-responding metabolic pathways in a minimally invasive manner.  The study is presented as a pilot-and-feasibility study in a relatively small group of adults.  Overall, the approach is exciting and the results should further advance the field of exercise metabolomics, especially when considered in light of previous findings.  Some unclear approaches and data presentation, however, made it difficult for this reviewer to fully embrace and understand the major findings.

Comments and Critiques

1.  It is stated that 33 metabolites were detected with “significant changes” but it is not clear how many total metabolites were detectable using the SESI-HRMS.

2.  It is not immediately clear how “significance” was evaluated—please be crystal clear as to how this was determined, over which time frame, etc.

3.  In several places, the term “intensity” is used but not well-defined. For each presentation of this term, in the paper and Supplemental files, define the term please.  As a related concern:  a heat map is presented but it is not really possible to understand how the investigators deemed “high” or “low” intensity or what the gradations are based on.

4.  Figure 1 is impossible to read with respect to the individual metabolite results in the small graphs. It would be more convincing to see the metabolites’ names, as currently presented, with harmonized indicators of increases or decreases in pathways (for instance, the glycine-associated metabolite section has no arrow, but the tryptophan and the TCA pathways do have arrows.  The individual metabolite changes including box plot results would be helpful to supply in supplemental (currently, only m/z defined metabolites are presented but they do not have a name, in supplemental).

5.  On line 160 there is reference to contrasting plasma and breath patterns but there is no context provided. What contrasting patterns?  There are no plasma results in this paper.

6.  Overall, the Discussion lacks depth with respect to commenting on how specific metabolites in breath during exercise in the current paper compare with what is published elsewhere (typically for plasma). What has been learned previously from plasma patterns from multiple labs, and do the current results complement those results, validate some findings, contrast with other findings?  Note that the paper would benefit (in the Introduction, and then Discussion as relevant) from consideration of key publications not currently cited: e.g., PMID: 31526287; PMID: 32869956; PMID: 27730694

7.  Related to Comment 6, reviewer found the Discussion to lack depth in terms of consideration of the physiological shifts during exercise and the potential etiologies of how the metabolite pathway patterns respond to these shifts.

Author Response

  1. It is stated that 33 metabolites were detected with “significant changes” but it is not clear how many total metabolites were detectable using the SESI-HRMS.

Response to comment 1 of reviewer 3

We thank the reviewer to point that out. Currently, it is unknown how many metabolites SESI-HRMS can detect. In our manuscript we focused on metabolites that are well described within the metabolic network. The 33 metabolites are the ones discussed closer in this manuscript. In this context, we adapted the result section to 32 significant changes between the start of the ramp and VO2max, since hydroxybutyrates’ significant change happens in the recovery phase.

  1. It is not immediately clear how “significance” was evaluated—please be crystal clear as to how this was determined, over which time frame, etc.

Response to comment 2 of reviewer 3

We thank the reviewer to point that out. We now adapted the methods part and the make it clearer.. “To determine significance in boxplots, q-values were calculated as mentioned above. Q-values of ≤ 0.001 were considered significant.”

  1. In several places, the term “intensity” is used but not well-defined. For each presentation of this term, in the paper and Supplemental files, define the term please.  As a related concern:  a heat map is presented but it is not really possible to understand how the investigators deemed “high” or “low” intensity or what the gradations are based on.

Response to comment 3 of reviewer 3

We thank the reviewer to point that out. We now define the intensity at the first point mentioned in the context of mass spectrometry. Furthermore we, now explain how the scale is chosen according to the range of observed values by clustergram Matlab algorithm.

  1. Figure 1 is impossible to read with respect to the individual metabolite results in the small graphs. It would be more convincing to see the metabolites’ names, as currently presented, with harmonized indicators of increases or decreases in pathways (for instance, the glycine-associated metabolite section has no arrow, but the tryptophan and the TCA pathways do have arrows.  The individual metabolite changes including box plot results would be helpful to supply in supplemental (currently, only m/z defined metabolites are presented but they do not have a name, in supplemental).

Response to comment 4 of reviewer 3

We thank the reviewer to point that out. The intension of the figure was to show the interplay of the different pathways over the whole continuous monitoring time of the exercise test. With the boxplot we intended to show the real data instead of the direction of changes. However, we adapted Table S1 and now include the changes of the different metabolites with arrows in this table. We therefore concluded to leave figure 1 as it was (with the option to zoom) and adapted table S1 accordingly. We also added the metabolite names in figure S4.

  1. On line 160 there is reference to contrasting plasma and breath patterns but there is no context provided. What contrasting patterns?  There are no plasma results in this paper.

Response to comment 5 of reviewer 3

We are not sure which reference the reviewer refers to, since the mentioned reference (20-22) all discuss plasma results.

  1. Overall, the Discussion lacks depth with respect to commenting on how specific metabolites in breath during exercise in the current paper compare with what is published elsewhere (typically for plasma). What has been learned previously from plasma patterns from multiple labs, and do the current results complement those results, validate some findings, contrast with other findings?  Note that the paper would benefit (in the Introduction, and then Discussion as relevant) from consideration of key publications not currently cited: e.g., PMID: 31526287; PMID: 32869956; PMID: 27730694

Response to comment 6 of reviewer 3

We thank the reviewer to point that out. Please refer to comment 5 of reviewer 1. “We are happy to take the table S1 into the main text if the editor wants us to do so. We have put this table into the supplement because we think that only few readers will be interested in the exact state of identification of the single metabolites.

We added the indication of “increased”, “continuous”, and “decreased” to the table S1

We agree that Figure 1 is too small to read in A4 format. Is it possible in this online journal to zoom the figures by e.g. clicking on the small version? Otherwise, we would need to cut the figure in several separate figures, containing only parts of the network, but we think this would reduce the clarity of the Figure and make it hard to read.”

Furthermore, we think the publications mentioned above, are too specific for the discussion of our manuscript, since they focus mainly on long-term changes in specific diseased populations. However, we now discuss the impact of our study in context to the literature in more detail.

  1. Related to Comment 6, reviewer found the Discussion to lack depth in terms of consideration of the physiological shifts during exercise and the potential etiologies of how the metabolite pathway patterns respond to these shifts.

Response to comment 7 of reviewer 3

We thank the reviewer to point that out. We agree with the reviewer and revised the discussion.